# Upregulation of LAMB1 via ERK/c-Jun Axis Promotes Gastric Cancer Growth and Motility

**DOI:** 10.3390/ijms22020626

**Published:** 2021-01-10

**Authors:** Hana Lee, Won-Jin Kim, Hyeon-Gu Kang, Jun-Ho Jang, Il Ju Choi, Kyung-Hee Chun, Seok-Jun Kim

**Affiliations:** 1Department of Integrative Biological Sciences & BK21 FOUR Educational Research Group for Age-Associated Disorder Control Technology, Chosun University, Gwangju 61452, Korea; hahanu808@chosun.kr (H.L.); wjsh003@naver.com (W.-J.K.); kang84562@nate.com (H.-G.K.); rndlvkf1@chosun.kr (J.-H.J.); 2Center for Gastric Cancer, National Cancer Center Research Institute, Goyang 10408, Korea; cij1224@ncc.re.kr; 3Department of Biochemistry & Molecular Biology, Yonsei University College of Medicine, 50 Yonsei-ro, Seodaemun-gu, Seoul 03722, Korea; khchun@yuhs.ac; 4Department of Biomedical Science, GARD Cohort Research Center, Chosun University, Gwangju 61452, Korea

**Keywords:** laminin subunit beta 1 (LAMB1), gastric cancer, cell proliferation, cell motility, ERK/c-Jun/LAMB1 axis

## Abstract

Gastric cancer is the fifth most common cancer worldwide with a poor survival rate. Therefore, it is important to identify predictive and prognostic biomarkers of gastric cancer. Laminin subunit beta 1 (LAMB1) is involved in attachment, migration, and organization during development, and its elevated expression has been associated with several cancers. However, the role and mechanism of LAMB1 in gastric cancer remains unknown. Here, we determined that LAMB1 is upregulated in gastric cancer tissues and contributes to cell growth and motility. Using a public database, we showed that LAMB1 expression was significantly upregulated in gastric cancer compared to normal tissues. LAMB1 was also found to be associated with poor prognosis in patients with gastric cancer. Overexpression of LAMB1 elevated cell proliferation, invasion, and migration; however, knockdown of LAMB1 decreased these effects in gastric cancer cells. U0126, an extracellular signal-regulated kinase (ERK) inhibitor, regulated the expression of LAMB1 in gastric cancer cells. Additionally, we showed that c-Jun directly binds to the LAMB1 promoter as a transcription factor and regulates its gene expression via the ERK pathway in gastric cancer cells. Therefore, our study indicates that LAMB1 promotes cell growth and motility via the ERK/c-Jun axis and is a potential biomarker and therapeutic target of gastric cancer.

## 1. Introduction

Gastric cancer is the most common contributor to the global burden of malignant tumors and responsible for a high mortality rate worldwide [1,2]. Despite the advancements in various cancer therapies, such as radiation therapy, immunotherapy, chemotherapy and surgery, the five-year relative cancer survival rate remains poor with approximately 20% [3]. There is an urgent need to develop the diagnosis and new therapies for gastric cancer, for which it is important to identify effective prognostic markers.

LAMB1, also known as Laminin β1, is composed of laminins and is present in most of the tissues [4]. Generally, LAMB plays a role in initiating cell assembly by combining with LAMC. Cell assembly plays an important role in cancerous cell invasion and metastasis [5]. It has been reported that LAMB1 is highly expressed in several invasive cancers [6]. Laminin is a representative extracellular matrix (ECM) glycoprotein, which is composed of large heterotrimeric protein complexes containing laminin subunit alpha, beta, and gamma (also called LAMA, LAMB, and LAMC), and can assemble 16 isoforms of laminins in humans [7]. Recently, laminin subunits, including LAMA, LAMB, and LAMC, have been shown to be involved in various cancer types affecting their oncogenic function, and thus, were revealed as potential therapeutic target markers based on data analysis [8,9,10,11]. In hepatocellular carcinoma (HCC), LAMB1 expression increases tumor progression during invasion via PDGF/La axis-mediated LAMB1 translation [12]. LAMB1 levels are high in the serum of colorectal cancer patients, so LAMB1 can be used as a potential serological biomarker [13]. However, the function and mechanism of LAMB1 remain undefined in gastric cancer. We hypothesized that LAMB1 might be involved in the oncogenic function and cell motility of invasion and migration of gastric cancer cells. In this study, we aimed to explore the effects of LAMB1 in gastric cancer.

The ERK pathway is a major trigger for the development of several cancers, including lung, colorectal, breast, liver, and gastric cancer. In several studies, laminin has been shown to be involved in the MAPK pathway, and MAPK/ERK signaling pathways regulate LAMB1 expression in HCC [12,14]. c-Jun has been found to be a transcription factor with oncogenic function in most cancers. The excessive expression of c-Jun plays an important role in various biological functions, such as apoptosis, proliferation, invasion, and migration [15,16,17]. It has been reported that the ERK pathway regulates c-Jun. c-Jun-induced matrix metalloproteinase-1 (MMP1) expression-promoted cell motility [18]. Moreover, c-Jun, a transcription factor, regulates laminin 1 (also called laminin 111-alpha 1, beta 1, and gamma 1)-induced cell growth [19]. We hypothesized that c-Jun is regulated via the ERK pathway and regulates the transcription of laminin and laminin subunits.

In this study, we demonstrate that LAMB1 expression is elevated in gastric cancer patients and that LAMB1 influences cell growth and motility in gastric cancer. Additionally, we demonstrate that LAMB1 expression is regulated via the ERK pathway. In addition, the transcription factor c-Jun can mediate LAMB1 transcription. Finally, we investigated the mechanism of LAMB1 function and revealed that it could serve as a potential biomarker and therapeutic target of gastric cancer.

## 2. Results

### 2.1. LAMB1 Is Upregulated in Gastric Cancer Patients in Public GEO Datasets

To identify the genes that are overexpressed in gastric cancer, we used the public database of four gene expression omnibus (GEO) datasets (GSE2685, GSE13861, GSE33651, and GSE63089) to analyze the differentially expressed genes (DEGs). Then, we searched for genes upregulated in gastric cancer compared to normal tissues and classified them in each public microarray dataset. The overall analysis pipeline is shown in Figure 1A. The methods used and detailed information of the GEO datasets are described in Table 1. GEO2R analysis using public GEO datasets generated Venn diagrams of 56 genes merging the overexpressed genes in four GSE datasets of gastric cancer tissues (Figure 1B). To identify the biological functions of these genes, we performed Kyoto Encyclopedia of Genes and Genomes (KEGG) pathway and gene ontology (GO) analyses KEGG pathway and GO annotation showed that the DEGs were most involved in various pathways, such as extracellular matrix (ECM)-receptor interaction, ECM organization, cell adhension, and structural constituents (Figure 1C and Appendix A). The data also showed that LAMB1 was involved in pathways, including ECM signaling and adhension, which mediated cell motility and progression in cancer (Table 1, Appendix A) [20]. Thus, we further explored the role of LAMB1 in this study. In addition, to confirm the distribution of LAMB1 in GSE datasets, we analyzed the relative gene expression. We showed that LAMB1 was located in the middle position of the upregulated genes (Figure 1D). The information about genes on the x-axis is described in Appendix A. These results indicate that LAMB1 is upregulated in gastric cancer based on the analysis of public GEO datasets.

### 2.2. LAMB1 Is Significantly Upregulated and Correlated with Risk of Poor Prognosis in Gastric Cancer

We reconfirmed the mRNA expression of LAMB1 using the publicly available GEO datasets of gastric cancer patients, which indicated significant overexpression of LAMB1 in gastric cancer (Figure 2A). We also investigated LAMB1 expression in gastric cancer patient tissues. Examination of LAMB1 expression in six patients with normal and tumor tissues for gastric cancer showed an increase in the expression of LAMB1 in tumor than normal tissues (Figure 2B). Kaplan–Meier analysis showed that overexpression of LAMB1 was associated with poor overall survival (OS), first progression (FP), and post-progression survival (PPS) of gastric cancer patients (Figure 2C and Appendix A). These results suggest that overexpression of LAMB1 might be a potential indicator for poor prognosis in gastric cancer.

### 2.3. LAMB1 Silencing Suppresses Proliferation, Invasion and Migration in Gastric Cancer Cells

We confirmed LAMB1 expression in normal gastric cancer cells and six gastric cancer cell lines (Appendix A). Both mRNA and protein expression levels of LAMB1 we upregulated in all gastric cancer cells, except in SNU-601 cells. To investigate the biological role of LAMB1 in gastric cancer, we generated AGS and MNN-28 cells for the knockdown of LAMB1 using designed siRNAs (Figure 3A). LAMB1 knockdown decreased cell proliferation in AGS and MNN-28 cells, which was confirmed using colony-formation assay but did not change the cell-cycle distribution (Figure 3B and Appendix A). To determine whether LAMB1 expression influences cell motility during invasion and migration in gastric cancer, we showed that LAMB1 knockdown decreased cell invasion and migration in AGS and MKN-28 cells using Transwell assay (Figure 3C,D). These data indicate that LAMB1 knockdown suppressed the biological role of proliferation, invasion, and migration in gastric cancer cells.

### 2.4. LAMB1 Enhances Proliferation, Invasion and Migration in Gastric Cancer Cells

LAMB1 overexpression in gastric cancer cells was examined in SNU-601 and SNU-668 cells transfected with mock vector or LAMB1 overexpression vector (Figure 4A). The colony-formation assay using the transfected SNU-601 and SNU-668 cells showed an increase in the number of colonies suggesting that LAMB1 overexpression induced cell proliferation (Figure 4B). Moreover, LAMB1 overexpression significantly enhanced cell invasion and migration in SNU-601 and SNU-668 cells, which was confirmed using Transwell assay (Figure 4C,D). These data suggest that LAMB1 overexpression induced cell proliferation, invasion, and migration in gastric cancer cells.

### 2.5. U0126 Inhibits LAMB1 Expression in Gastric Cancer Cells

In hepatocellular carcinoma (HCC), LAMB1 is known to upregulate cell signaling via the ERK and Akt pathways [12]. The ERK pathway is dominantly regulated by LAMB1 expression in HCC. To identify the effect of the ERK pathway on LAMB1 expression in gastric cancer, we treated AGS and MKN-28 cells with the ERK pathway inhibitor, U0126. Western blot analysis of AGS and MKN-28 cells treated with U0126 for 24 h showed that the protein expression of LAMB1 was decreased upon U0126 treatment compared to DMSO treatment (Figure 5A). U0126 decreased the expression of LAMB1, thereby inhibiting its biological function of cell proliferation, invasion, and migration in gastric cancer (Figure 5B–D. We examined whether the rescue of LAMB1 expression can induce the proliferation, invasion, and migration of cancer cells. We pretreated 10 μM U0126 for 24 h and then transfected with LAMB1 overexpression vector in AGS and MKN-28 cells (Figure 5A). The recovery of LAMB1 expression in U0126-treated AGS and MKN-28 cells significantly increased cell proliferation, invasion, and migration (Figure 5B–D). These data indicate that LAMB1 expression is regulated via the ERK pathway and affects their biological function in gastric cancer.

### 2.6. c-Jun Binds to LAMB1 Promoter Activate Transcription in Gastric Cancer Cells

We screened the LAMB1 promoter to identify the transcription factor binding sites using PROMO 3.0. We found potential transcription factors that could bind to the proximal LAMB1 promoter, including c-Jun, C/EBPβ, RXRα, and YY1 (Figure 6A). To investigate the effects of transcription factor-regulated mRNA levels of LAMB1, we generated AGS cells for knockdown using various siRNAs (Appendix A). This result showed that c-Jun knockdown effectively inhibited the mRNA expression of LAMB1 in AGS cells. Moreover, the c-Jun-binding motif is present in the LAMB1 promoter (<2000 bp) region, and thus, we designed two potential binding sites (−1403 bp to −1225 bp and −1046 bp to −851 bp). ChIP assays showed that c-Jun directly binds to LAMB1 promoter in gastric cancer cells (Figure 6B). In the above results, U0126-treated gastric cancer cells exhibited decreased LAMB1 expression (Figure 5A). It has been reported that the expression of c-Jun is regulated via the ERK pathway in various cancers, including gastric cancer, mediating various cellular processes of cell growth, invasion, and migration [21]. We showed that U0126, an ERK inhibitor, decreased the protein expression of c-Jun and LAMB1, but reactivation of LAMB1 expression through the transfection of LAMB1 overexpression vector in U0126-treated AGS and MKN-28 cells did not influence c-Jun expression (Figure 6C and Appendix A). Using the public GEO datasets, we showed a correlation between JUN (gene name of c-Jun) and LAMB1 gene expression (Figure 6D). Moreover, to determine whether knockdown of c-Jun inhibited the mRNA and protein expression of LAMB1 in gastric cancer, we treated AGS and MKN-28 cells with JUN-specific siRNAs and found that c-Jun knockdown decreased LAMB1 expression at both mRNA and protein levels in AGS and MKN-28 cells (Figure 6E). These results suggest that c-Jun directly binds to the LAMB1 promoter and regulates the transcriptional gene expression of LAMB1 via the ERK pathway in gastric cancer cells. Moreover, ERK/c-Jun/LAMB1 pathway may serve as a novel therapeutic target in gastric cancer (Figure 7).

## 3. Discussion

Gastric cancer is an important contributor to the global burden of cancer for the past several decades [22]. Diagnostic and surgical techniques have slightly improved the survival rate in gastric cancer patients. Although many potential biomarkers for gastric cancer have been reported, limited therapeutic effects [23]. It is important to identify therapeutic biomarkers of gastric cancer. Recently, it was reported that epithelial-mesenchymal transition (EMT) progression is required for the initiation of metastasis and tumor formation [24]. Extracellular matrix (ECM) components, such as laminin, are important regulators of EMT, as well as tumor invasion and metastasis [20]. Extracellular matrix (ECM) is an important component of the tumor microenvironment, which influences critical biochemical and biomechanical effects of cancer development, progression, metastasis, and immune function [25,26]. The ECM has not only as of the scaffold upon but mediates critical biochemical and biomechanical roles [25]. Upregulated or abnormally activated ECM proteins promote cell proliferation, invasion, and migration in cancer [27]. The interactions between various cell surface growth receptors and ECM components can influence the organization of the cytoskeleton and intracellular signaling [26]. Thus, ECM components represent the potential biomarkers for cancer diagnosis and prognosis.

Laminin activates various signal transduction pathways. Laminin involves tumor cell attachment and proliferation laminin and induces invasion [28]. In metastatic cells, laminin promotes activation of matrix metalloproteinase-2 (MMP2), which has a role in tumor cell metastasis [29]. Recently, a study has reported that laminin chains have overexpression and poor prognosis in various cancer patients [30,31,32]. LAMB1 gene was overexpressed in several invasive cancers [6]. Reanalysis using GEO2R of in public GSE datasets showed that the overexpressed genes, including LAMB1, are involved in pathways such as cell adhesion and ECM interaction. We focused on LAMB1 as a potential biomarker and its cellular mechanisms in gastric cancer. In this study, public datasets have demonstrated that overexpression of LAMB1 in gastric cancer promotes tumor growth and is a marker of poor prognosis. Knockdown of LAMB1 in vitro inhibited cancer cell growth and motility, whereas overexpression of LAMB1 significantly enhanced cell proliferation, invasion, and migration of gastric cancer cells. These results suggest that LAMB1 functions in tumor growth and motility in gastric cancer.

The ECM-integrin receptor interaction activates the MAPK pathway [33,34]. The MAPK pathway is essential for ECM determined cell survival. The MAPK/ERK signaling pathway plays a crucial role in gastric cancer, such as tumorigenesis and progression [35]. Moreover, the ERK pathway regulates the invasion, migration, and metastasis progression of malignant cancer. Interaction between laminin and cell surface receptors is an important process in signal transduction pathways and is related to kinase-phosphatase cascade stimuli gene expression and cellular function [36,37]. Recently, studies have reported that the interaction between integrin and ECM promotes the intracellular pathway of phosphorylated FAK and the ERK signaling in human gastric cancer cells [38]. LAMB1 expression is regulated via PDGF signaling, which activates the MAPK/ERK and PI3K/Akt pathways in HCC, and the MAPK/ERK pathway efficiently mediates further activation of LAMB1 express than PI3K/Akt pathway [12]. We used the inhibitor of U0126 (ERK inhibitor) and LY294002 (Akt inhibitor). Similar to the above, LAMB1 was regulated on gastric cancer cells through the ERK pathway compared to the Akt pathway. We found that the ERK pathway regulates LAMB1 expression levels in gastric cancer, demonstrating the regulation of LAMB1 via the ERK pathway, thereby inducing cancer cell growth and motility. Inhibition of the ERK pathway is also a potential therapeutic target marker in gastric cancer with LAMB1 overexpression.

We showed that LAMB1 is upregulated at both gene and protein expression levels in gastric cancer tissues and cell lines. In HCC, the signaling pathway regulated LAMB1 translation during EMT, and cytoplasmic laminin affects the translation of LAMB1 in cells [12]. However, little is known about the transcription factor or regulator of LAMB1 in cancer. We focused on identifying a transcription factor regulating LAMB1 expression in gastric cancer. It has been reported that the transcription factor of the AP-1 complex binds and regulates LAMB1 gene expression upon binding to its promoter in F9 mouse embryonal cells and LAMC2 in HT29 human colon cancer cells [39,40]. AP-1 complexes are mediated via the MAP kinase family, including ERKs and p38 [41]. c-Jun can promote the expression of genes associated with ECM components and mediate cell growth and invasion in cancer [18,42]. Additionally, c-Jun regulates the expression of laminin [19]. However, the transcriptional role of c-Jun in regulating LAMB1 gene expression in gastric cancer has not been studied yet. In this study, we observed a strong correlation between LAMB1 and c-Jun gene expression. In addition, we found that c-Jun can bind LAMB1 promoter regions and function as a transcription factor to regulate LAMB1 gene expression in gastric cancer, which needs to be explored further to elucidate its function. The transcriptional activation of LAMB1 by c-Jun in gastric cancer, which needs to be explored further.

In ECM remodeling in cancer, increased specific protein from ECM turnover is released into the blood [43]. ECM proteins can be potentially used as a biomarker for early and efficient detection. In early-stage of liver cancer, elevated LAMB1 mediated the increased secretion of laminin 111 (Ln-111) [33]. The overexpressed laminin 111 interacts with its cell membrane receptor, which further mediates cellular signaling, and promotes tumor growth and metastasis. Although, we have shown the tissue LAMB1 expression is upregulated. Moreover, we need to identify in the further study whether LAMB1 promotes the secretion and expression of laminin 111 in gastric cancer. We believe that the overexpression of LAMB1 may induce laminin overexpression, thereby mediating cancer progression and metastasis, and can serve as a critical biomarker of gastric cancer.

## 4. Materials and Methods

### 4.1. Cell Lines and Patient Tissues

Human gastric cancer cell lines, including AGS, MKN-28, YCC-2, SNU-216, SNU-601, and SNU-668, were purchased from the Korean Cell Line Bank (Laboratory of Cell Biology, Cancer Research Center and Cancer Research Institute, Seoul National University College of Medicine, Seoul, Korea). Human normal gastric epithelial cell line (GES-1) was obtained from Yonsei Cancer Center (Seoul, Korea). All cells were cultured in RPMI-1640 medium (Welgene, Gyeongsan-si, Korea) containing 5% fetal bovine serum (Corning Costar, Corning, NY, USA) and 1% antibiotic-antimycotic (Gibco, Waltham, MA, USA) in a 37 °C incubator supplied with 5% atmospheric CO_2_. Two pairs of 2-mm-sized biopsy specimens were obtained from 52 patients with gastric adenocarcinoma during diagnostic endoscopic submucosal dissection. Immediately after biopsy, these tissue samples were frozen in liquid nitrogen in a deep-freezer at −70 °C until further use. All participants provided written informed consent. All experimental procedures were approved by the Institutional Review Board of the National Cancer Center (approval number, NCCNCS-09-215; approval data, 02/26/2009).

### 4.2. Plasmid Infection and Transient Transfection

LAMB1 cDNA clones were provided by the Korea Human Gene Bank (KHGB). The human LAMB1 construct was cloned into the pCMV-3Tag-1A plasmid in BamH1/Xho1 restriction enzyme sites to generate pCMV-LAMB1. Primer sequences for LAMB1 with BamH1/Xho1 enzyme site are 5′-AATACGGATCCATGGGGCTTCAGTTGCT-3′ (forward, with BamH1) and 5′-ACGAGCTCGAGTTACAAGCATGTGCTATACA-3′ (reverse, with Xho1). The sequences of human siRNAs purchased from Genolution Inc (Genolution, Seoul, Korea)., and siRNA sequences used in this study are as follows: LAMB1_siRNA#1, 5′-GAGAUAACCUUCUGGAUUCUU-3′ (forward) and 5′-GAAUCCAGAAGGUUAUUAUCUCUU-3′ (reverse); LAMB1_siRNA#2, 5′-GGAUUUCUACCAUGAUUUAUU-3′ (forward) and 5′-UAAAUCAUGGUAGAUUUAUU-3′ (reverse); c-Jun_siRNA#1, 5′-GAGCUGGAGCGCCUGAUAAUU-3′ (forward) and 5′-UUAUCAGGCGCUCCAGCUCUU-3′ (reverse); c-Jun_siRNA#2, 5′-GAGCGGACCUUAUGGCUACUU-3′ (forward) and 5′-GUAGCCAUAAGGUCCGCUCUU-3′ (reverse); C/EBPβ_siRNA, 5′-CCAAGAAGACCGUGGACAAUU-3′ (forward) and 5′-UACUCGGCCGGCUUCUUGCUU-3′ (reverse); RXRα_siRNA, 5′-GGGAGAAGGUCUAUGCGUCUU-3′ (forward) and 5′-GACGCAUAGACCUUCUCCCUU-3′ (reverse); and YY1_siRNA, 5′-GGAUAACUCGGCCAUGAGAUU-3′ (forward) and 5′-UCUCAUGGCCGAGUUAUCCUU-3′ (reverse). The cells were transfected using lipofectamine 2000 (Invitrogen, Waltham, MA, USA) for plasmid DNA transfection and lipofectamine RNAiMAX (Invitrogen, Waltham, MA, USA) for siRNA infection following the manufacturer’s protocol. AGS, MKN-28, SNU-601 and SNU-668 cells were transfected with 2 μg of plasmid DNA at a 60 mm cell culture dish. Moreover, AGS and MKN-28 cells were transfected with 20 μM of LAMB1_siRNA#1, #2 or 30 μM siRNA (c-Jun, C/EBPβ, RXRα, and YY1) at a 60 mm cell culture dish. The transfected cells were incubated for 48 h in a 37 °C incubator supplied with 5% atmospheric CO_2_.

### 4.3. Total RNA Isolation and Reverse Transcriptase–Polymerase Chain Reaction (RT–PCR)

Total RNA was isolated using RNAiso Plus reagent (TaKaRa Bio, Kusatsu, Japan), and cDNA was synthesized using cDNA Master Mix (ToYoBo, Ōsaka, Japan), according to the manufacturer’s instructions. Then, PCR was performed by using HiPi Plus 5 × PCR Master Mix (Elpis Biotech, Daejeon, Korea). The primer sequences used in this study were as follows: LAMB1, 5′-AGGTTGGAGCTGCCTCAGTA-3′ (forward) and 5′-ACACTCCCTGGAAACAGTGG-3′ (reverse); JUN, 5′-CCCCAAGATCCTGAAACAGA-3′ (forward) and 5′-CCGTTGCTGGACTGGATTAT-3′ (reverse); CEBPB, 5′-CAAGAAGCCGGCCGAGTAC-3′ (forward) and 5′-TTGTCCACGGTCTTCTTGGC-3′ (reverse); RXRA, 5′-CCTTTCTCGGTCATCAGCTC-3′ (forward) and 5′-TGTCAATCAGGCAGTCCTTG-3′ (reverse); YY1, 5′-GGATAACTCGGCCATGAGAA-3′ (forward) and 5′-GGTTGTTTTTGGCCTTAGCA-3′ (reverse); GAPDH, 5′-TGCACCACCAACTGCTTAG-3′ (forward) and 5′-GGATGCAGGGATGATGTTC-3′ (reverse). PCR products were separated using 1% agarose gel electrophoresis and visualized using RedSafe nucleic acid staining solution (iNtRON Biotechnology, seongnam, Korea).

### 4.4. Protein Preparation and Western Blotting

The cells were lysed to extract the proteins using RIPA buffer. After incubation at 4 °C for 35 min, each sample was centrifuged at 13,200 rpm at 4 °C for 25 min. The supernatant contained the protein. Protein concentration was measured using the bovine serum albumin (BSA) protein assay (Thermo Fisher Scientific, Waltham, MA, USA). A defined quantity of total protein was electrophoresed using 10–8% SDS–PAGE gels, and the resolved protein bands were transferred to PVDF membranes. Each membrane was blocked using 5% skimmed milk prepared in 0.05% Tween-20 with 1 × PBS (PBST) for 1 h 30 min at RT. After the reaction of blocking, each membrane was incubated at 4 °C overnight with primary antibody diluted (1:1000) in 5% BSA prepared in PBST buffer. Each membrane was incubated with diluted (1:5000) secondary antibody for 1 h 30 min at RT. The bands were visualized through chemiluminescence (Bio-Rad, Seoul, Korea).

### 4.5. Antibodies

The monoclonal antibodies used were anti-laminin β-1, anti-ERK, anti-p-ERK, and anti-β-actin (Santa Cruz Biotechnology), and the polyclonal antibodies used were anti-c-Jun (Santa Cruz Biotechnology) and glyceraldehyde-3-phosphate dehydrogenase (GAPDH, Bioworld Technology, St Louis Park, MN, USA).

### 4.6. Colony-Formation Assay

Transfected cells were plated in a 35 mm culture dish (1500 cells). During incubation, the medium was changed after every 3 days. After 10 days of incubation, colonies were washed with cold 1 × PBS and fixed with 1% glutaraldehyde for 10 min. Colonies were stained with 0.5% crystal violet for 10 min. Thereafter, the colonies formed in each dish were counted. Each experiment was performed in triplicates.

### 4.7. Invasion and Migration Assay

Transfected cells were plated in FBS-free medium at 2 × 10^4^ cells in the upper chamber of a Transwell (Corning Costar) on a filter coated with 0.5 mg/mL collagen type I (BD Biosciences, Franklin Lakes, NJ, USA) to perform the migration assay, and 1:15 diluted Matrigel (BD Biosciences) coated filter to perform the invasion assay. RPMI 1640 medium with 10% FBS and 1% antibiotics was added 1 mL to the lower chamber, followed by incubation for 20 h in a 37 °C incubator supplied with 5% atmospheric CO_2_. Cell invasion and migration were visualized and quantified by H&E staining. For quantification, cells were counted from three randomly selected areas using wide-filed microscopy.

### 4.8. Cell Cycle Analysis

Transfected cells were incubated for 48 h at 37 °C. After incubation, cells were washed with 1 × PBS and fixed in 5 mL of 75% ethanol at −20 °C overnight. After fixation, cells were washed twice with cold 1 × PBS and dispersed in a staining solution containing 50 μg/mL RNase A and 50 μg/mL PI solution in PBS for 15 min at RT. Cell cycle distribution was assessed after PI staining using the CytoFLEX flow cytometer (Beckman Coulter, Brea, CA, USA).

### 4.9. Chromatin Immunoprecipitation (ChIP) Assay

AGS and MKN-28 cells of gastric cancer were cultured in a 150 mm culture dish (2 × 10^5^ cells). After 2 days of incubation, medium containing cells were treated with 1% formaldehyde for cross-linking. ChIP assay was performed using the Pierse Agarose ChIP Kit (Thermo Fisher Scientific), according to the manufacturer’s instructions. Anti-c-Jun and normal rabbit lgG were used to immunoprecipitate DNA-containing complexes. Anti-c-Jun was the same that antibody used in Western blotting. Anti-rabbit lgG was contained in the ChIP kit and used as a negative control. Primers were designed with LAMB1 promoter binding sites: Primer 1, 5′-CTTCTCTGGGCCTTATTTCG-3′ (forward) and 5′-CTGCTACCCTTAGCAATGGA-3′ (reverse), which was amplified region of 204-bp; primer 2, 5′-GGAGAATCGTCGAGATGAGC-3′ (forward) and 5′-CTGGGCAACAAGAGCAAAAC-3′ (reverse), which was amplified region of 213-bp. Moreover, PCR was performed by using HiPi Plus 5 × PCR Master Mix (Elpis Biotech, Daejeon, Korea).

### 4.10. Data Accession and Analysis

The data generated in this publication have been deposited to NCBI’s Gene Expression Omnibus (https://www.ncbi.nlm.nih.gov/geo) with GEO series accession numbers GSE2685, GSE13861, GSE33651, and GSE63089. These data were normalized using GEO2R. Genes with logFC > 0.7 were considered as differentially expressed genes (DEGs). As most data values were log2-transformed in advance, we reprocessed the data accordingly. The distribution of overexpressed genes in gastric cancer tissue is shown. Moreover, we confirm the overall upregulated gene expression locus in public datasets. The gene expression in normal tissues was calculated as one (relative expression > 1 means overexpression in gastric cancer tissues).

### 4.11. Statistical Analysis

Data analyses were performed using GraphPad Prism5 software (GraphPad, San Diego, CA, USA). Statistical analyses were performed using Student’s *t*-test. Overall, survival, first progression, and post-progression survival of gastric cancer patients were analyzed using Kaplan–Meier analysis (https://kmplot.com/analysis) and the data were generated using the gene symbol LAMB1 (Affy ID: 201505_at and 211651_s_at). Gene ontology (GO) enrichment and Kyoto Encyclopedia of Genes and Genomes (KEGG) pathway analyses were performed using the Database for Annotation, Visualization and Integrated Discovery (DAVID; david.ncifcrf.gov). *p*-value < 0.05 was considered to be statistically significant. Moreover, all data presented as mean ± SEM.

## 5. Conclusions

Our results revealed that LAMB1 is significantly overexpressed in gastric cancer tissues and exerts the biological function of promoting tumor growth and cell invasion and migration of gastric cancer cells. In this study, we showed that ERK/c-Jun regulates LAMB1 expression and that LAMB1 is a potential therapeutic target for developing the treatment of gastric cancer.

## Figures and Tables

**Figure 1 ijms-22-00626-f001:**
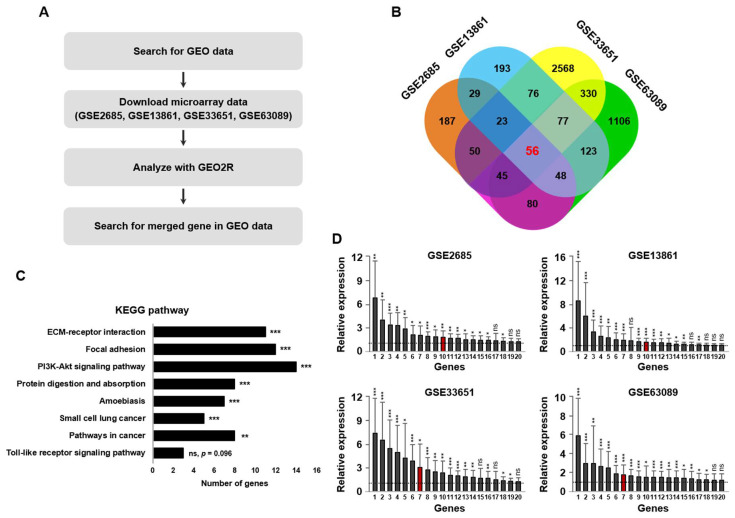
Laminin subunit beta 1 (LAMB1) gene expression is upregulated in public datasets of gastric cancer patients. (**A**) Schematic presentation of the used gene expression omnibus (GEO) datasets. GEO series accession numbers were GSE2685, GSE13861, GSE33651, and GSE63089. (**B**) Venn diagrams showing upregulated genes (logFC > 0.7, logFC > 0 means overexpression in gastric cancer tissues) in gastric cancer tissues after applying GEO2R analysis. Numbers in red indicate merged genes in a public database of the four GSE datasets used for the analysis. (**C**) Kyoto Encyclopedia of Genes and Genomes (KEGG) pathway analysis of upregulated genes in a public database of GSE datasets used. The arrangement in order of low *p*-value is shown here. (**D**) Overall, the upregulated gene expression locus in gastric cancer tissue is presented. Red bar indicates LAMB1 locus. The x-axis showed the genes from GSE datasets. * *p* < 0.05; ** *p* < 0.01; *** *p* < 0.001, ns: not significant.

**Figure 2 ijms-22-00626-f002:**
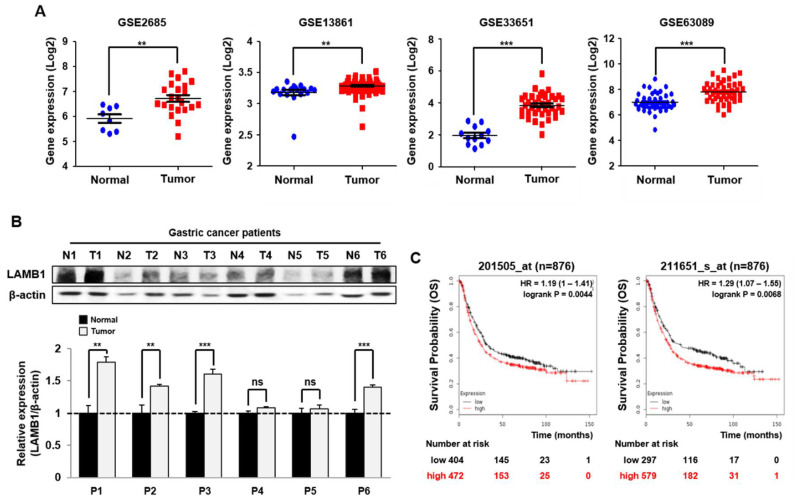
Overexpression of LAMB1 is correlated with an increased risk of gastric cancer. (**A**) Examination of LAMB1 gene expression in normal and gastric cancer tissues using a public database of GSE datasets. (**B**) Protein expression of LAMB1 in six patients of gastric normal and tumor tissues using Western blot analysis. Relative expression of LAMB1 was analyzed by the ImageJ program. (**C**) Kaplan–Meier plots of the overall survival related to LAMB1 expression in a public database of gastric cancer patients. The data are presented as mean ±SEM. ** *p* < 0.01; *** *p* < 0.001, ns: not significant.

**Figure 3 ijms-22-00626-f003:**
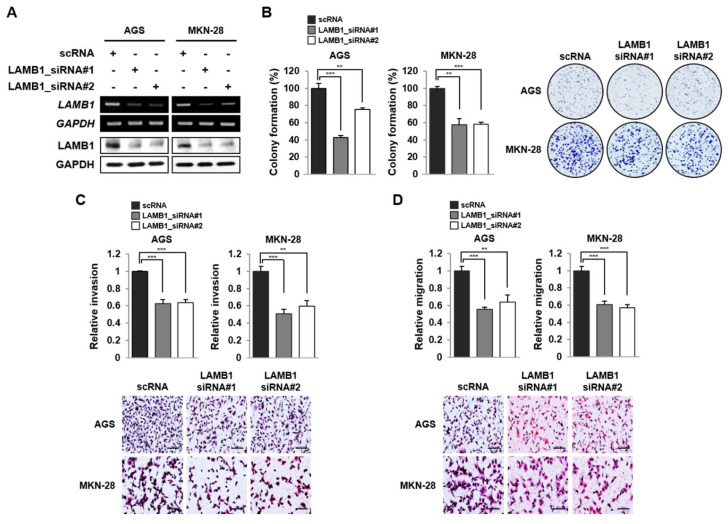
LAMB1 knockdown inhibits proliferation, invasion, and migration of AGS and MKN-28 cells. (**A**) mRNA and protein expression levels of LAMB1 in AGS and MKN-28 cells upon scrambled siRNA (scRNA) or LAMB1 siRNA#1, #2 transfection. (**B**) Colony-formation assay showing cell proliferation of transfected AGS and MKN-28 cells upon LAMB1 knockdown. (**C**) Cell invasion and (**D**) migration assays using transfected AGS and MKN-28 cells. Image magnification × 200; scale bar, 50 μm. The data are presented as mean ± SEM (*n* = 3). ** *p* < 0.01; *** *p* < 0.001.

**Figure 4 ijms-22-00626-f004:**
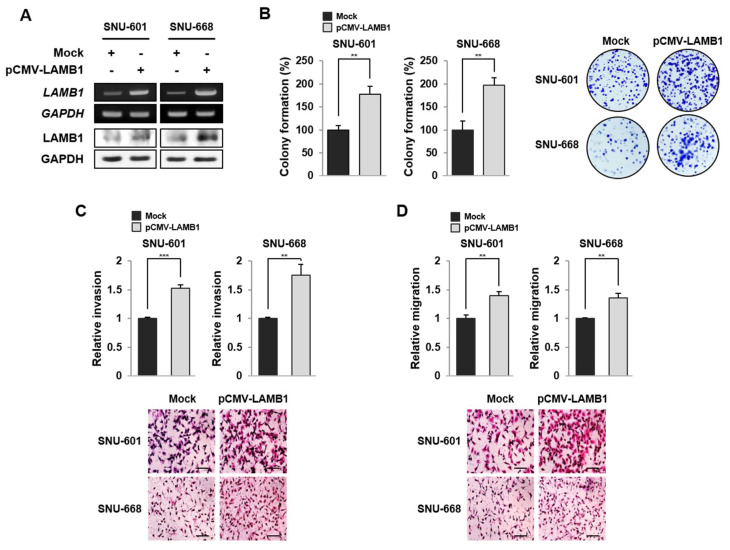
LAMB1 overexpression promotes proliferation, invasion, and migration of SNU-601 and SNU-668 cells. (**A**) mRNA and protein expression levels of LAMB1 in SNU-601 and SNU-668 cells transfected with pCMV-3Tag-1A (Mock, empty vector) or pCMV-3Tag-1A-LAMB1 (pCMV-LAMB1, LAMB1 overexpression vector). (**B**) Colony-formation assay showing cell proliferation of transfected SNU-601 and SNU-668 cells with LAMB1 overexpression. (**C**) Cell invasion and (**D**) migration assays using transfected SNU-601 and SNU-668 cells. Image magnification × 200; scale bar, 50 μm. The data are presented as mean ± SEM (*n* = 3). ** *p* < 0.01; *** *p* < 0.001.

**Figure 5 ijms-22-00626-f005:**
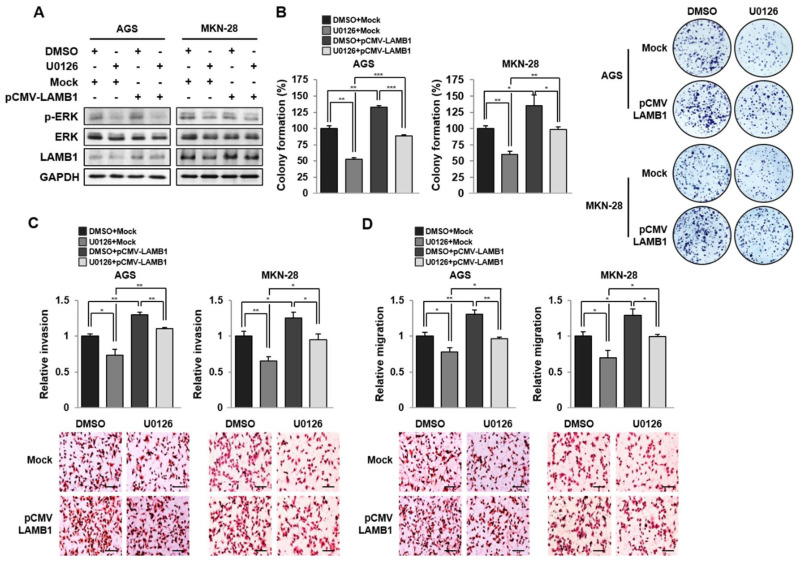
Extracellular signal-regulated kinase (ERK) pathway inhibition reduces LAMB1 expression in AGS and MKN-28 cells. (**A**) Protein expression level of LAMB1 in AGS and MKN-28 cells transfected with pCMV-3Tag-1A (Mock, empty vector) or pCMV-3Tag-1A-LAMB1 (pCMV-LAMB1, LAMB1 overexpression vector) after pretreatment with 10 μM U0126 for 24 h. (**B**) Colony-formation assay showing cell proliferation of transfected AGS and MKN-28 cells with LAMB1 overexpression after pretreatment with 10 μM U0126 for 24 h. (**C**) Cell invasion and (**D**) migration assays using transfected AGS and MKN-28 cells after pretreatment with 10 μM U0126 for 24 h. Image magnification × 200; scale bar, 50 μm. The data are presented as mean ± SEM (*n* = 3). * *p* < 0.05; ** *p* < 0.01; *** *p* < 0.001.

**Figure 6 ijms-22-00626-f006:**
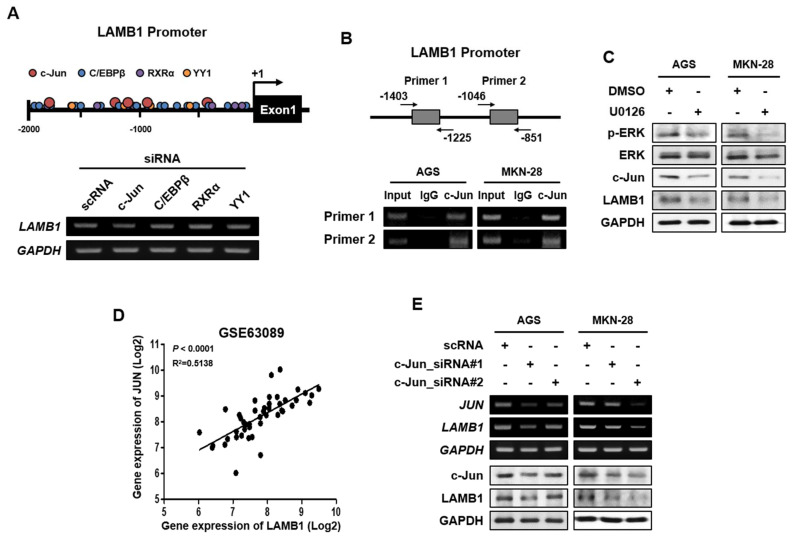
c-Jun binds to LAMB1 promoter and mediates its transcription in AGS and MKN-28 cells. (**A**) Schematic presentation of transcription factor binding sites on LAMB1 promoter. Prediction program for promoter binding sites using PROMO 3.0. In addition, mRNA expression of LAMB1 in AGS cells upon transfection with scrambled siRNA (scRNA) or various siRNA (used siRNA of c-Jun, C/EBPβ, RXRα and YY1). (**B**) Chip assay revealed that anti-c-Jun antibody binds to LAMB1 promoter. (**C**) Protein expression of c-Jun in AGS and MKN-28 cells upon pretreatment with 10 μM U0126 (ERK inhibitor). DMSO was used as a control. (**D**) Correlation between gene expression of c-Jun and LAMB1 in a public database of GSE63089. The correlation analysis was performed using GraphPad Prism5 software. (**E**) mRNA and protein expression of c-Jun in AGS and MKN-28 cells transfected with scrambled siRNA (scRNA) or c-Jun siRNA#1 and #2.

**Figure 7 ijms-22-00626-f007:**
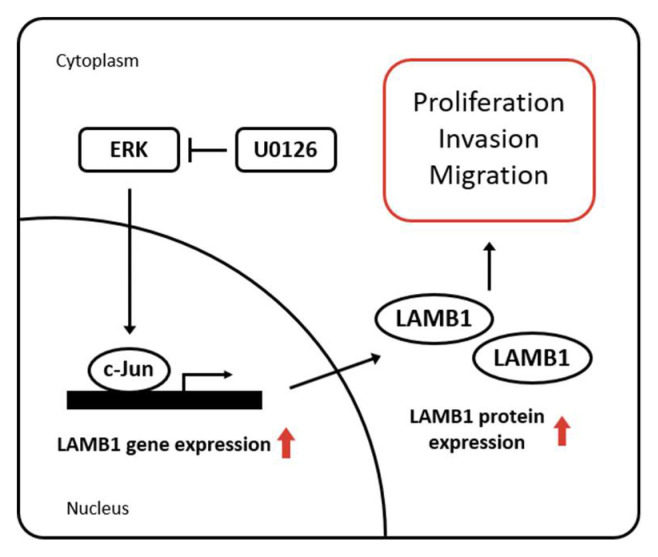
Schematic model illustrating the role of LAMB1 in cell growth and motility through ERK/c-Jun in gastric cancer. ERK/c-Jun axis elevates LAMB1 expression in gastric cancer. c-Jun directly binds to the LAMB1 promoter and regulates the gene and protein expression levels of LAMB1. Therefore, overexpressed LAMB1 promotes cell proliferation, invasion, and migration in gastric cancer cells (red arrow).

**Table 1 ijms-22-00626-t001:** Relevant information retrieved from gastric cancer datasets of four GSE microarrays.

GEO Accession	Contributors, Year	Country	Platform	Total Samples	Cancer Tissue	Normal Tissue
GSE2685	Hippo, Y. et al., 2002	Japan	GPL80	30	22	8
GSE13861	Cho, J. et al., 2011	USA	GPL6884	90	71	19
GSE33651	Park, D. et al., 2011	South Korea	GPL2895	52	40	12
GSE63089	Zhang, X. et al., 2014	China	GPL5175	90	45	45

## Data Availability

The data presented in this study are available on request from the corresponding author.

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
