# Peer review of "Upregulation of LAMB1 via ERK/c-Jun Axis Promotes Gastric Cancer Growth and Motility"

_ijms, 2021, doi:10.3390/ijms22020626_

Round 1

Reviewer 1 Report

Novel biomarkers for gastric cancer prognosis and therapeutic targets are needed. This manuscript describes an investigation into the role of LAMB1 and ERK/c-Jun in gastric cancer pathogenesis. The rationale for conducting these experiments is nicely described, and the approach is sound. The series of experiments follows a logical progression and the appropriate controls were performed.

The discussion centers on data interpretation and could be expanded to further describe the potential implications of this work. What further studies would be needed to assess the utility LAMB1 as a gastric cancer prognostic marker or as a therapeutic target? How might these results be incorporated into clinical management of patients with gastric cancer in the future?

There are several grammatical problems with text of this manuscript with recurring themes being an overuse of the word "therefore" and when to use "the," as well as instances where abbreviations are not spelled out. I have listed many of them below in an effort to improve the grammar:

Line 30: spell out ERK

Line 44: remove "Therefore" from this sentence

Line 50: change cancer to "cancers"

Line 58: remove ",and thus," from this sentence

Line 59: remove "Therefore" from this sentence

Line 61: remove "Therefore" from this sentence

Line 62: Begin this sentence with "The ERK pathway..."

Lines 67, 160, 162, 245, 246, 252, 254 : Insert "the" before "ERK pathway"

Line 70: remove "Therefore" from this sentence

Lines 72-76: This paragraph should be in the present tense, so change "demonstrated" to "demonstrate."

Line 75: Remove "In this study" and replace with "Finally"

Line 80: Spell out the first instance of GEO and DEG

Line 88: Spell out the first instance of ECM

Line 94: Insert "the" before LAMB1

Line 115: replace "risk factor" with "indicator"

Line 161: remove the word "majorly"

Line 183: "factors" should not be pleural

Line 222: change "since" to "for the past"

Line 223: Remove "the" from the beginning of this sentence

Line 224, 230: remove "Therefore" from this sentence

Line 233: Remove "A study reported that"

Line 235: Remove the comma after "pathways"

Line 239: Replace "caused" with "was associated with"

Line 247: Remove "the" before "cell surface"

Line 255: Remove "The" and start the sentence with "Inhibition"

Line 258: Remove "both, the"

Line 263: Insert "the" before "transcription factor"

Line 272: Remove the s in "details"

Line 273: Remove "the" before "early stage"

Lines 274-278: The phrase "in the supernatant" is confusing and redundant and should be removed from each sentence.

Line 275: Change the wording of this sentence to "with it's cell membrane receptor, which further..."

Figure S2: Change "Provability to "Probability" on the y axis. Change "prognostics" to "prognosis" in the figure legend.

Author Response

Author’s response

Dear editor

We have carefully revised our manuscript after reading the comments provided by the reviewers. Please find our manuscript “ijms-1047840_ Upregulation of LAMB1 via ERK/c-Jun axis promotes gastric cancer growth and motility”. First, we would like to thank the editor and reviewers for the constructive and helpful comments and suggestions. According to the comments, we have revised the manuscript, which mainly including the experiment results and rewriting of each section. Following are a point-by-point response to the reviewers’ comments. Once again, we appreciate the editor and reviewers very much for comments on our paper.

Looking forward to hearing from you. Thank you and best regards.

Best Regards

Yours Sincerely,

                                                            Seok-Jun Kim

Reviewer #1

Novel biomarkers for gastric cancer prognosis and therapeutic targets are needed. This manuscript describes an investigation into the role of LAMB1 and ERK/c-Jun in gastric cancer pathogenesis. The rationale for conducting these experiments is nicely described, and the approach is sound. The series of experiments follows a logical progression and the appropriate controls were performed. The discussion centers on data interpretation and could be expanded to further describe the potential implications of this work.

  1. What further studies would be needed to assess the utility LAMB1 as a gastric cancer prognostic marker or as a therapeutic target? How might these results be incorporated into clinical management of patients with gastric cancer in the future?

Answer: Thank you for your kind comment. As an additional study, we plan to do the following: First, LAMB1 is intended to confirm changes in its expression in the blood of gastric cancer patients. Through the article (Lin Q et al, Proteomics. 2015 Nov;15(22):3905-20) showed that (LAMB1) as a potential serological biomarker for colorectal cancer. Therefore, we are expecting the same effect in stomach cancer. After that, we intend to apply it to the development of diagnostic devices for stomach cancer based on the relevant results.

  1. There are several grammatical problems with text of this manuscript with recurring themes being an overuse of the word "therefore" and when to use "the," as well as instances where abbreviations are not spelled out. I have listed many of them below in an effort to improve the grammar:

Answer: Thank you for your kind comment. According to reviewer’s comments, we were change to the all section and rewritten. As followed, we organized the English editing company and grammar. Also, we added the relevant certification form with review files.

Line 30: spell out ERK

Answer: Line 31-32: rewrite “an extracellular signal-regulated kinase (ERK) inhibitor”

Line 44: remove "Therefore" from this sentence

Answer: Line 44: remove “Therefore” and rewrite “…approximately 20% [3]. There is an…”

Line 50: change cancer to "cancers"

Answer: Line 50: rewrite “invasive cancers”

Line 58: remove ",and thus," from this sentence

Answer: Line 58: remove “,and thus” and rewrite “…patients, so LAMB1 can…”

Line 59: remove "Therefore" from this sentence

Answer: Line 58: remove “Therefore” and rewrite “…cancer. We hypothesized…”

Line 61: remove "Therefore" from this sentence

Answer: Line 58: remove “Therefore” and rewrite “…cancer cells. In this study, …”

Line 62: Begin this sentence with "The ERK pathway..."

Answer: Line 62: rewrite “The ERK pathway”

Lines 67, 160, 162, 245, 246, 252, 254 : Insert "the" before "ERK pathway"

Answer: Line 67, 161-163, 252-264: insert “the” and rewrite “the ERK pathway”

Line 70: remove "Therefore" from this sentence

 Answer: Line 62: remove “Therefore” and rewrite “…cell growth [19]. We hypothesized…”

Lines 72-76: This paragraph should be in the present tense, so change "demonstrated" to "demonstrate."

Answer: Line 72-73: rewrite “we demonstrate…”

Line 75: Remove "In this study" and replace with "Finally"

Answer: Line 75: remove “In this study” and rewrite “Finally, we…”

Line 80: Spell out the first instance of GEO and DEG

Answer: Line 80-81: rewrite “gene expression omnibus (GEO)” and “differentially expressed genes (DEGs)”

Line 88: Spell out the first instance of ECM

Answer: Line 88: rewrite “extracellular matrix (ECM)”

Line 94: Insert "the" before LAMB1

Answer: Line 93: we change this sentence and rewrite “We showed that LAMB1 was located in the middle position of the upregulated genes”

Line 115: replace "risk factor" with "indicator"

Answer: Line 116: remove “risk factor” and rewrite “indicator”

Line 161: remove the word "majorly"

Answer: Line 162: we change “majorly” to “dominantly”. Because we chose ERK pathway instead of Akt pathway based on previously reported study.

Line 183: "factors" should not be pleural

Answer: Line 184: change “factors” to “factor”

Line 222: change "since" to "for the past"

Answer: Line 223: change “since” to “for the past”

Line 223: Remove "the" from the beginning of this sentence

Answer: Line 224: remove “the” and rewrite “Diagnostic and…”

Line 224, 230: remove "Therefore" from this sentence

Answer: Line 226, 233: remove “therefore” and rewrite “…effects [23]. It is…” and “…roles [25]. Upregulated…”

Line 233: Remove "A study reported that"

Answer: Line 242: remove “A study reported that” and rewrite “LAMB1 gene…”

Line 235: Remove the comma after "pathways"

Answer: Line 244: change “…pathways, …” to “…pathways…”

Line 239: Replace "caused" with "was associated with"

Answer: Line 244: change “…pathways, …” to “…pathways…”

Line 247: Remove "the" before "cell surface"

 Answer: Line: we remove this sentence. Because we change and rewrite discussion section.

Line 255: Remove "The" and start the sentence with "Inhibition"

Answer: Line 265: remove “The” and rewrite “Inhibition of…”

Line 258: Remove "both, the"

Answer: Line 271: remove “both, the” and rewrite “…levels in gastric…”

Line 263: Insert "the" before "transcription factor"

Answer: Line 270: insert “the” and rewrite “…that the transcription factor…”

Line 272: Remove the s in "details"

Answer: Line 280: remove “details” and rewrite “…its function.”

Line 273: Remove "the" before "early stage"

Answer: Line 285: remove “the” and rewrite “In early stage…”

Lines 274-278: The phrase "in the supernatant" is confusing and redundant and should be removed from each sentence.

Answer: Line 285-287: remove “in the supernatant” from this sentence

Line 275: Change the wording of this sentence to "with it's cell membrane receptor, which further..."

 Answer: Line 286: change “with the receptor on cell membrane, which further…” to “with it’s cell membrane receptor, which further…”

Figure S2: Change "Provability to "Probability" on the y axis. Change "prognostics" to "prognosis" in the figure legend.

Answer: Figure 2C and S2: change “Provability” to “Probability” on the y axis. And Figure S2 legend: change “prognostics” to “prognosis”.

Reviewer 2 Report

The authors performed an interesting functional study of LAMB1 in the setting of gastric cancer. They developed a logical research design to explore the effect of LAMB1 on the growth and invasion of gastric cancer. The results were clearly displayed in informative and detailed figures, and the results supported the conclusions of the study. The study is well within the scope of the journal and will be of interest to the readership, as well as physicians, pathologists, and scientists who work on gastrointestinal neoplasms and the mechanisms of invasion. I have the following comment:

The manuscript was fairly easy to read; however, there were several grammatical errors and awkwardly worded sentences. The authors need to carefully read the manuscript and address these errors in English grammar to improve the readability of the paper. Collaboration with a native English speaker or company familiar with scientific writing could be useful if deemed necessary by the authors.

Author Response

Author’s response

Dear editor

We have carefully revised our manuscript after reading the comments provided by the reviewers. Please find our manuscript “ijms-1047840_ Upregulation of LAMB1 via ERK/c-Jun axis promotes gastric cancer growth and motility”. First, we would like to thank the editor and reviewers for the constructive and helpful comments and suggestions. According to the comments, we have revised the manuscript, which mainly including the experiment results and rewriting of each section. Following are a point-by-point response to the reviewers’ comments. Once again, we appreciate the editor and reviewers very much for comments on our paper.

Looking forward to hearing from you. Thank you and best regards.

Best Regards

Yours Sincerely,

                                                            Seok-Jun Kim

Reviewer #2

The authors performed an interesting functional study of LAMB1 in the setting of gastric cancer. They developed a logical research design to explore the effect of LAMB1 on the growth and invasion of gastric cancer. The results were clearly displayed in informative and detailed figures, and the results supported the conclusions of the study. The study is well within the scope of the journal and will be of interest to the readership, as well as physicians, pathologists, and scientists who work on gastrointestinal neoplasms and the mechanisms of invasion. I have the following comment:

  1. The manuscript was fairly easy to read; however, there were several grammatical errors and awkwardly worded sentences. The authors need to carefully read the manuscript and address these errors in English grammar to improve the readability of the paper. Collaboration with a native English speaker or company familiar with scientific writing could be useful if deemed necessary by the authors.

Answer: Thank you for your kind comment. According to reviewer’s comments, we were change to the all section and rewritten. As followed, we organized the English editing company and grammar. Afterthat, we added the relevant certification form with review files.

Reviewer 3 Report

The study entitled “Upregulation of LAMB1 via ERK/c-Jun axis promotes  2

gastric cancer growth and motility” is somehow interesting and repertoire of methods used is reasonable, but spelling, grammar and quality of language makes this manuscript hard to follow and understand.

Sentences in lines 89-93 in Results should be rewrittnen as they provide the same information. Referencing is also needed.

This sentence “Upon  assessing  the  distribution  of  the  overall  upregulation  of  gene  93

expression  in  gastric  cancer  compared  to  normal  tissues,  we  found  that  LAMB1  locus  was” also should be corrected.

Generally Results section is full of grammar errors and should be proofread before publication. Materials and methods section is in turn too little detailed and missing some vital information.

Figure 1, panel D is hard to understand.  X axis is not labelled and it is not explained how these results were obtained.

How the authors obtained normal gastric tissue?

Western blots in Figure 2 are not convincing, specifically as it is not described anywhere how the results of relative expression were obtained. And for WB methods – what abs were used? Primary secondary? Clone numbers?

Thermo Fisher – is spelled this way…

Methods 4.2 title states that lentiviruses were used…  only plasmid transfection is described there.

Using the term “invasion” when only transwell migration assay is performed seems to be overstated

Sentence in line 168 is hard to understand: “We transfected  U0126-treated  AGS and MKN-28 cells  with LAMB1  overexpression vector before transfection (Figure 5A)”

Figure 6C is mission relative expression calculation since we do not know finally whether ERK phosphorylation is inhibited by U0126.

It would be beneficial to evaluate whether ERK pathway is overactivated in gastric cancer samples and cell lines.

The same here in discussion line 252: “…and MAPK/ERK pathway efficiently mediates further activation of LAMB1 express than PI3K/Akt pathway [12]

Discussion section is in my opinion too short and specifically is not elaborating properly on the subject of ERK overactivation/(overexpression?).

Author Response

Author’s response

Dear editor

We have carefully revised our manuscript after reading the comments provided by the reviewers. Please find our manuscript “ijms-1047840_ Upregulation of LAMB1 via ERK/c-Jun axis promotes gastric cancer growth and motility”. First, we would like to thank the editor and reviewers for the constructive and helpful comments and suggestions. According to the comments, we have revised the manuscript, which mainly including the experiment results and rewriting of each section. Following are a point-by-point response to the reviewers’ comments. Once again, we appreciate the editor and reviewers very much for comments on our paper.

Looking forward to hearing from you. Thank you and best regards.

Best Regards

Yours Sincerely,

                                                            Seok-Jun Kim

Reviewer #3

The study entitled “Upregulation of LAMB1 via ERK/c-Jun axis promotes gastric cancer growth and motility” is somehow interesting and repertoire of methods used is reasonable, but spelling, grammar and quality of language makes this manuscript hard to follow and understand.

  1. Sentences in lines 89-93 in Results should be rewrittnen as they provide the same information. Referencing is also needed.

Answer: Thank you for the comment. We insert reference and rewrite this sentence as “KEGG pathway and GO annotation showed that the DEGs were most involved in various pathways, such as extracellular matrix (ECM)-receptor interaction, ECM organization, cell adhesion, and structural constituents (Figure 1C and S1). The data also showed that LAMB1 was involved in pathways, including ECM signaling and adhesion, which mediated cell motility and progression in cancer (Table 1 and S2) [20].”.

  1. This sentence “Upon assessing the distribution of the overall upregulation of gene expression in gastric cancer compared to normal tissues, we found that LAMB1 locus was” also should be corrected.

Answer: Thank you for the comment. We rewrite this sentence as “In addition, to confirm the distribution of LAMB1 in GSE datasets, we analyzed the relative gene expression. We showed that LAMB1 was located in the middle position of the upregulated genes (Figure 1D). The information about genes on the x axis is described in Table S3.”.

  1. Generally Results section is full of grammar errors and should be proofread before publication. Materials and methods section is in turn too little detailed and missing some vital information.

Answer: Thank you for the comment. We confirm and change gramma errors. And we add the detail and missing information of materials and methods. Ex) we write the missing antibody information (b-actin) and some regent.

  1. Figure 1, panel D is hard to understand. X axis is not labelled and it is not explained how these results were obtained.

Answer: Thank you for the comment. We confirm and inset information of X axis. And we add the detail gene list in Table S3. And we insert the in materials and methods.

  1. How the authors obtained normal gastric tissue?

Answer: Thank you for the comment. One of the authors, Il-Ju Choi as a clinician, obtained normal tissue present near gastric and gastric cancer tissues using endoscopy. Relevant evidence is referred to in the material and method sections.

  1. Western blots in Figure 2 are not convincing, specifically as it is not described anywhere how the results of relative expression were obtained. And for WB methods – what abs were used? Primary secondary? Clone numbers?

Answer: Thank you for the comment. We used ImageJ program for calculation. We insert information in Figure 2 legend.

  1. Thermo Fisher – is spelled this way…

Answer: Thank you for the comment. We change “Thermo Fisher Scientific”.

  1. Methods 4.2 title states that lentiviruses were used… only plasmid transfection is described there.

Answer: Thank you for the comment. Also, we apologize for the confusion through the wrong description. Therefore, we were changed title to plasmids construction and transfection

  1. Using the term “invasion” when only transwell migration assay is performed seems to be overstated

Answer: Thank you for the comment. In general, the invasion assay uses two experimental methods (1. Wound healing assay 2. Transwell invasion assay). In this experiment, we were used transwell invasion assay. This experimental method is different from the transwell migration assay. Also, those different experimental methods describe in materials and methods section.

  1. Sentence in line 168 is hard to understand: “We transfected U0126-treated AGS and MKN-28 cells with LAMB1 overexpression vector before transfection (Figure 5A)”

Answer: Thank you for the comment. We rewrite for easy understand as “We pretreated 10 μM U0126 for 24 h and then transfected with LAMB1 overexpression vector in AGS and MKN-28 cells (Figure 5A).”.

  1. Figure 6C is mission relative expression calculation since we do not know finally whether ERK phosphorylation is inhibited by U0126.

Answer: Thank you for the comment. Although we do not calculate relative expression, phosphorylated ERK expression is efficiently regulated in U0126 treated gastric cancer cells.

  1. It would be beneficial to evaluate whether ERK pathway is overactivated in gastric cancer samples and cell lines.

Answer: We appreciate the reviewer’s comment. Actually, we didn’t confirm ERK activity in gastric cancer cell lines. Through the many reports showed that the high activity of ERK in gastric cancer, therefore it is considered worth checking. However, in this study, we were focus on LAMB1 expression, so we would like to apply it to additional research.

  1. The same here in discussion line 252: “…and MAPK/ERK pathway efficiently mediates further activation of LAMB1 express than PI3K/Akt pathway [12]

Answer: Thank you for the comment. We chose ERK pathway instead of Akt pathway based on previously reported study. We emphasize that ERK pathway is regulate the LAMB1 expression. So, we write similar sentence in discussion.

  1. Discussion section is in my opinion too short and specifically is not elaborating properly on the subject of ERK overactivation/(overexpression?).

Answer: Thank you for the comment. We were rewrite discussion section according to your comments.

Round 2

Reviewer 3 Report

Overall, the authors significantly corrected the manuscript. However, it still contains some awkwardly sounding sentences and phrases, i.e.:

Line 91 “signaling and adhension” – should be “adhesion”?

Line 127 “(Figure  S3). Both, mRNA and protein expression levels of LAMB1 we  upregulated in all gastric cancer” 

Line 133 “These  data  indicate  that  LAMB1  knockdown suppressed the biological role of proliferation, invasion, and migration in gastric”

Line 169 “We pretreated 10 μM U0126 for 24 h and then transfected with LAMB1 overexpression vector in AGS and MKN -28 cells” – should be “We pretreated AGS and MKN -28 cells with 10 μM U0126 for 24 h and then….”?

Line 173 “regulated via the ERK pathway and affects their biological function in gastric cancer.”  - “their”?

Line 373 “using wide-filed microscopy”

Line 386 “Anti-rabbit lgG was contained the ChIP kit and used as negative control”